# MASTERING VISUAL CONTINUOUS CONTROL: IMPROVED DATA-AUGMENTED REINFORCEMENT LEARNING

## ABSTRACT

We present DrQ-v2, a model-free reinforcement learning (RL) algorithm for visual continuous control. DrQ-v2 builds on DrQ, an off-policy actor-critic approach that uses data augmentation to learn directly from pixels. We introduce several improvements that yield state-of-the-art results on the DeepMind Control Suite. Notably, DrQ-v2 is able to solve complex humanoid locomotion tasks directly from pixel observations, previously unattained by model-free RL. DrQ-v2 is conceptually simple, easy to implement, and provides significantly better computational footprint compared to prior work, with the majority of tasks taking just 8 hours to train on a single GPU. Finally, DrQ-v2's implementation is publicly released to provide RL practitioners with a strong and computationally efficient baseline.

## 1 INTRODUCTION

Creating sample-efficient continuous control methods that observe high-dimensional images has been a long standing challenge in reinforcement learning (RL) . Over the last three years, the RL community has made significant headway on this problem, improving sample-efficiency significantly. The key insight to solving visual control is the learning of better low-dimensional representations, either through autoencoders (Yarats et al., 2019; Finn et al., 2015), variational inference (Hafner et al., 2018; 2019; Lee et al., 2019), contrastive learning (Srinivas et al., 2020; Yarats et al., 2021a), self-prediction (Schwarzer et al., 2020b), or data augmentations (Yarats et al., 2021b; Laskin et al., 2020). However, current state-of-the-art model-free methods are still limited in three ways. First, they are unable to solve the more challenging visual control problems such as quadruped and humanoid locomotion. Second, they often require significant computational resources, i.e. lengthy training times using distributed multi-GPU infrastructure. Lastly, it is often unclear how different design choices affect overall system performance.

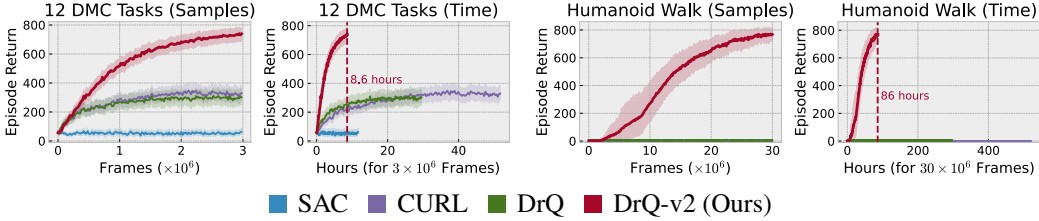

Figure 1: DrQ-v2 demonstrates significantly better sample efficiency and computational footprint compared to state-of-the-art model-free methods for visual continuous control while being conceptually simple and easy to implement. (**Left two**) Average performance results across 12 challenging tasks from the DeepMind Control Suite (the set of tasks can be seen in Figure 8). (**Right two**) Performance on the *Humanoid Walk* task from visual input, previously unsolved by model-free methods. In both cases we report sample complexity and wall-clock time axes for evaluation, with time being measured on a single GPU machine and using official implementations for each method.

In this paper we present DrQ-v2, a simple model-free algorithm that builds on the idea of using data augmentations (Yarats et al., 2021b; Laskin et al., 2020) to solve hard visual control problems. Most notably, it is the first model-free method that solves complex humanoid tasks directly from pixels. Compared to previous state-of-the-art model-free methods, DrQ-v2 provides significant improvements in sample efficiency across tasks from the DeepMind Control Suite (Tassa et al., 2018). Conceptually simple, DrQ-v2 is also computationally efficient, which allows solving most tasks in DeepMind Control Suite in just 8 hours on a single GPU (see Figure 1). Recently, a model-based method, DreamerV2 (Hafner et al., 2020) was also shown to solve visual continuous control problems and it was first to solve the humanoid locomotion problem from pixels. While our model-free DrQ-v2 matches DreamerV2 in terms sample efficiency and performance, it does so $4\times$ faster in terms of wall-clock time to train. We believe this makes DrQ-v2 a more accessible approach to support research in visual continuous control and it reinforces the question on whether model-free or model-based is the more suitable approach to solve this type of tasks.

DrQ-v2, which is detailed in Section 3, improves upon DrQ (Yarats et al., 2021b) by making several algorithmic changes: (i) switching the base RL algorithm from SAC (Haarnoja et al., 2018a) to DDPG (Lillicrap et al., 2015a) with clipped double Q-learning from TD3 (Fujimoto et al., 2018), (ii) this allows us straightforwardly incorporating multi-step return, (iii) adding bilinear interpolation to the random shift image augmentation, (iv) introducing an exploration schedule, (v) selecting better hyper-parameters including a larger capacity of the replay buffer. A careful ablation study of these design choices is presented in Section 4.4. Furthermore, we re-examine the original implementation of DrQ and identify several computational bottlenecks such as replay buffer management, data augmentation processing, batch size, and frequency of learning updates (see Section 3.2). To remedy these, we have developed a new implementation that both achieves better performance and trains around 3.5 times faster with respect to wall-clock time than the previous implementation on the same hardware with an increase in environment frame throughput (FPS) from 28 to 96 (i.e., it takes $10^6/96/3600 \approx 2.9$ hours to train for 1M environment steps). DrQ-v2's implementation is available at https://anonymous.4open.science/r/drqv2.

## 2 BACKGROUND

### 2.1 REINFORCEMENT LEARNING FROM IMAGES

We formulate image-based control as an infinite-horizon Markov Decision Process (MDP) (Bellman, 1957). Generally, in such a setting, an image rendering of the system is not sufficient to perfectly describe the system's underlying state. To this end and per common practice (Mnih et al., 2013), we approximate the current state of the system by stacking three consecutive prior observations. With this in mind, such MDP can be described as a tuple $(\mathcal{X}, \mathcal{A}, P, R, \gamma, d_0)$, where $\mathcal{X}$ is the state space (a three-stack of image observations), $\mathcal{A}$ is the action space, $P : \mathcal{X} \times \mathcal{A} \to \Delta(\mathcal{X})$ is the transition function[1] that defines a probability distribution over the next state given the current state and action, $R : \mathcal{X} \times \mathcal{A} \to [0, 1]$ is the reward function, $\gamma \in [0, 1)$ is a discount factor, and $d_0 \in \Delta(\mathcal{X})$ is the distribution of the initial state $\boldsymbol{x}_0$. The goal is to find a policy $\pi : \mathcal{X} \to \Delta(\mathcal{A})$ that maximizes the expected discounted sum of rewards $\mathbb{E}_\pi[\sum_{t=0}^{\infty} \gamma^t r_t]$, where $\boldsymbol{x}_0 \sim d_0$, and $\forall t$ we have $\boldsymbol{a}_t \sim \pi(\cdot|\boldsymbol{x}_t)$, $\boldsymbol{x}_{t+1} \sim P(\cdot|\boldsymbol{x}_t, \boldsymbol{a}_t)$, and $r_t = R(\boldsymbol{x}_t, \boldsymbol{a}_t)$.

### 2.2 DEEP DETERMINISTIC POLICY GRADIENT

Deep Deterministic Policy Gradient (DDPG) (Lillicrap et al., 2015a) is an actor-critic algorithm for continuous control that concurrently learns a Q-function $Q_\theta$ and a deterministic policy $\pi_\phi$. For this, DDPG uses Q-learning (Watkins and Dayan, 1992) to learn $Q_\theta$ by minimizing the one-step Bellman residual $J_\theta(\mathcal{D}) = \mathbb{E}_{(\boldsymbol{x}_t, \boldsymbol{a}_t, r_t, \boldsymbol{x}_{t+1}) \sim \mathcal{D}}[(Q_\theta(\boldsymbol{x}_t, \boldsymbol{a}_t) - r_t - \gamma Q_{\bar\theta}(\boldsymbol{x}_{t+1}, \pi_\phi(\boldsymbol{x}_{t+1})))^2]$. The policy $\pi_\phi$ is learned by employing Deterministic Policy Gradient (DPG) (Silver et al., 2014) and maximizing $J_\phi(\mathcal{D}) = \mathbb{E}_{\boldsymbol{x}_t \sim \mathcal{D}}[Q_\theta(\boldsymbol{x}_t, \pi_\phi(\boldsymbol{x}_t))]$, so $\pi_\phi(\boldsymbol{x}_t)$ approximates $\operatorname{argmax}_{\boldsymbol{a}} Q_\theta(\boldsymbol{x}_t, \boldsymbol{a})$. Here, $\mathcal{D}$ is a replay buffer of environment transitions and $\bar\theta$ is an exponential moving average of the weights. DDPG is amenable to incorporate $n$-step returns (Watkins, 1989; eng and Williams, 1996) when estimating TD error beyond a single step (Barth-Maron et al., 2018). In practice, $n$-step returns allow for faster

---

[1]Here, $\Delta(\mathcal{X})$ denotes a distribution over the state space $\mathcal{X}$.

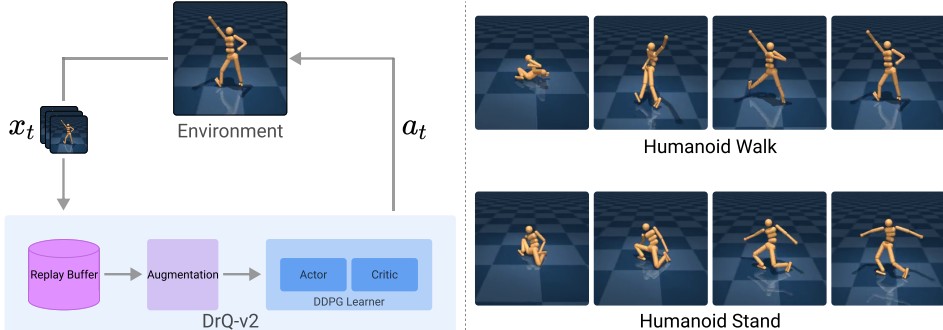

Figure 2: (Left): DrQ-v2 is an off-policy actor-critic algorithm for image-based RL. It alleviates encoder overfitting by applying random shift augmentation to pixel observations sampled from the replay buffer. (Right): Examples of walking and standing behaviors learned by DrQ-v2 for a complex humanoid agent from DMC (Tassa et al., 2018) with 21 and 54 dimensional action and state spaces, respectively. DrQ-v2 does not have access to the internal state of the environment, only observing three consecutive pixel frames at a time. Despite this imperfect observational channel, our agent still manages to solve the tasks. To the best of our knowledge, this is the first successful demonstration by a model-free method, using pixel-based inputs of these tasks.

reward propagation and has been previously used in policy gradient and Q-learning methods (Mnih et al., 2016b; Barth-Maron et al., 2018; Hessel et al., 2017).

### 2.3 DATA AUGMENTATION IN REINFORCEMENT LEARNING

Recently, it has been shown that data augmentation techniques, commonplace in Computer Vision, are also important for achieving the state-of-the-art performance in image-based RL (Yarats et al., 2021b; Laskin et al., 2020). For example, the state-of-the-art algorithm for visual RL, DrQ (Yarats et al., 2021b) builds on top of Soft Actor-Critic (Haarnoja et al., 2018a), a model-free actor-critic algorithm, by adding a convolutional encoder and data augmentation in the form of random shifts. The use of such data augmentations now forms an essential component of several recent visual RL algorithms (Srinivas et al., 2020; Raileanu et al., 2020; Yarats et al., 2021a; Stooke et al., 2020; Hansen and Wang, 2021; Schwarzer et al., 2020b).

## 3 DRQ-V2: IMPROVED DATA-AUGMENTED REINFORCEMENT LEARNING

In this section, we describe DrQ-v2, a simple model-free actor-critic RL algorithm for image-based continuous control, that builds upon DrQ.

### 3.1 ALGORITHMIC DETAILS

**Image Augmentation** As in DrQ we apply random shifts image augmentation to pixel observations of the environment. In the settings of visual continuous control by DMC, this augmentation can be instantiated by first padding each side of $84 \times 84$ observation rendering by $4$ pixels (by repeating boundary pixels), and then selecting a random $84 \times 84$ crop, yielding the original image shifted by $\pm 4$ pixels. We also find it useful to apply bilinear interpolation on top of the shifted image (i.e, we replace each pixel value with the average of the four nearest pixel values). In our experiments, this modification provides an additional performance boost across the board.

**Image Encoder** The augmented image observation is then embedded into a low-dimensional latent vector by applying a convolutional encoder. We use the same encoder architecture as in DrQ, which first was introduced introduced in SAC-AE (Yarats et al., 2019). This process can be succinctly summarized as $\boldsymbol{h} = f_\xi(\mathrm{aug}(\boldsymbol{x}))$, where $f_\xi$ is the encoder, $\mathrm{aug}$ is the random shifts augmentation, and $\boldsymbol{x}$ is the original image observation.

**Actor-Critic Algorithm** We use DDPG (Lillicrap et al., 2015a) as a backbone actor-critic RL algorithm and, similarly to Barth-Maron et al. (2018), augment it with $n$-step returns to estimate TD error. This results into faster reward propagation and overall learning progress (Mnih et al., 2016a).

While some methods (Hafner et al., 2020) employ more sophisticated techniques such as TD($\lambda$) or Retrace($\lambda$) (Munos et al., 2016), they are often computationally demanding when $n$ is large. We find that using simple $n$-step returns, without an importance sampling correction, strikes a good balance between performance and efficiency. We also employ clipped double Q-learning (Fujimoto et al., 2018) to reduce overestimation bias in the target value. Practically, this requires training two Q-functions $Q_{\theta_1}$ and $Q_{\theta_2}$. For this, we sample a mini-batch of transitions $\tau = (\boldsymbol{x}_t, \boldsymbol{a}_t, r_{t:t+n-1}, \boldsymbol{x}_{t+n})$ from the replay buffer $\mathcal{D}$ and compute the following two losses:

$$\mathcal{L}_{\theta_k, \xi}(\mathcal{D}) = \mathbb{E}_{\tau \sim \mathcal{D}}\big[(Q_{\theta_k}(\boldsymbol{h}_t, \boldsymbol{a}_t) - y)^2\big] \quad \forall k \in \{1, 2\}, \tag{1}$$

with the TD target $y$ defined as:

$$y = \sum_{i=0}^{n-1} \gamma^i r_{t+i} + \gamma^n \min_{k=1,2} Q_{\bar{\theta}_k}(\boldsymbol{h}_{t+n}, \boldsymbol{a}_{t+n}),$$

where $\boldsymbol{h}_t = f_\xi(\text{aug}(\boldsymbol{x}_t))$, $\boldsymbol{h}_{t+n} = f_\xi(\text{aug}(\boldsymbol{x}_{t+n}))$, $\boldsymbol{a}_{t+n} = \pi_\phi(\boldsymbol{h}_{t+n}) + \epsilon$, $\bar{\theta}_1$ and $\bar{\theta}_2$ are the slow-moving weights for the Q target networks. We note, that in contrast to DrQ, we do not employ a target network for the encoder $f_\xi$ and always use the most recent weights $\xi$ to embed $\boldsymbol{x}_t$ and $\boldsymbol{x}_{t+n}$. The exploration noise $\epsilon$ is sampled from $\text{clip}(\mathcal{N}(0, \sigma^2), -c, c)$ similar to TD3 (Fujimoto et al., 2018), with the exception of decaying $\sigma$, which we describe below. Finally, we train the deterministic actor $\pi_\phi$ using DPG with the following loss:

$$\mathcal{L}_\phi(\mathcal{D}) = -\mathbb{E}_{\boldsymbol{x}_t \sim \mathcal{D}}\big[\min_{k=1,2} Q_{\theta_k}(\boldsymbol{h}_t, \boldsymbol{a}_t)\big], \tag{2}$$

where $\boldsymbol{h}_t = f_\xi(\text{aug}(\boldsymbol{x}_t))$, $\boldsymbol{a}_t = \pi_\phi(\boldsymbol{h}_t) + \epsilon$, and $\epsilon \sim \text{clip}(\mathcal{N}(0, \sigma^2), -c, c)$. Similar to DrQ, we do not use actor's gradients to update the encoder's parameters $\xi$.

**Scheduled Exploration Noise** Empirically, we observe that it is helpful to have different levels of exploration at different stages of learning. At the beginning of training we want the agent to be more stochastic and explore the environment more effectively, while at the later stages of training, when the agent has already identified promising behaviors, it is better to be more deterministic and master those behaviors. Similar to Amos et al. (2020), we instantiate this idea by using linear decay $\sigma(t)$ for the variance $\sigma^2$ of the exploration noise defined as:

$$\sigma(t) = \sigma_{\text{init}} + (1 - \min(\frac{t}{T}, 1))(\sigma_{\text{final}} - \sigma_{\text{init}}), \tag{3}$$

where $\sigma_{\text{init}}$ and $\sigma_{\text{final}}$ are the initial and final values for standard deviation, and $T$ is the decay horizon.

**Key Hyper-Parameters** We conduct an extensive hyper-parameter search and identify several hyper-parameter changes compared to DrQ. The three most important hyper-parameters are: (i) the size of the replay buffer, (ii) mini-batch size, and (iii) learning rate. Specifically, we use a 10 times larger replay buffer than DrQ. We also use a smaller mini-batch size of 256 without any noticeable performance degradation. This is in contrast to CURL (Srinivas et al., 2020) and DrQ (Yarats et al., 2021b) that both use a larger batch size of 512 to attain more stable training in the expense of computational efficiency. Finally, we find that using smaller learning rate of $1 \times 10^{-4}$, rather than DrQ's learning rate of $1 \times 10^{-3}$, results into more stable training without any loss in learning speed.

## 3.2 IMPLEMENTATION DETAILS

**Faster Image Augmentation** We replace DrQ's random shifts augmentation (i.e., `kornia.augmentation.RandomCrop`) by a custom implementation that uses flow-field image sampling provided in PyTorch (i.e., `grid_sample`). This is done for two reasons. First, we noticed that Kornia's implementation does not fully utilize GPU pipelining since it has some intermediate CPU to GPU data transferring which breaks the computational flow. Second, using `grid_sample` allows straightforward addition of bilinear interpolation. Our custom random shifts augmentation improves training throughput by a factor of 2.

**Faster Replay Buffer** Another computational bottleneck of DrQ was the replay buffer. The specific implementation had poor memory management which resulted in slow CPU to GPU data transfer, which also restricted the number of image-based transitions that could be stored. We reimplemented the replay buffer to address these issues which led to a ten-fold increase in storage capacity and faster data transfer. More details are available in our open-source release. We note that the improved training speed of DrQ-v2 was key to solving humanoid tasks as it enabled much faster experimentation.

## 4 EXPERIMENTS

In this section we provide empirical evaluation of DrQ-v2 on an extensive set of visual continuous control tasks from DMC (Tassa et al., 2018). We first present comparison to prior methods, both model-free and model-based, in terms of sample efficiency and wall-clock time. We then present a large scale ablation study that guided the final version of DrQ-v2.

### 4.1 SETUP

**Environments** We consider a set of MuJoCo tasks (Todorov et al., 2012) provided by DMC (Tassa et al., 2018), a widely used benchmark for continous control. DMC offers environments of various difficulty, ranging from the simple control problems such as the single degree of freedom (DOF) pendulum and cartpool, to the control of complex multi-joint bodies such as the humanoid (21 DOF). We consider learning from pixels. In this setting, environment observations are stacks of 3 consecutive RGB images of size $84 \times 84$, stacked along the channel dimension to enable inference of dynamic information like velocity and acceleration. In total, we consider 24 different tasks, which we group into three buckets, *easy*, *medium*, and *hard*, according to the sample complexity to reach near-optimal performance (see Appendix B). Our motivation for this is to encourage RL practitioners to focus on the medium and hard tasks and stop using the easy tasks for evaluation, as they are mostly solved at this point and may no longer provide any valuable signal in comparing different methods.

**Training Details** For all tasks in the suite an episode corresponds to 1000 steps, where a per-step reward is in the unit interval $[0, 1]$. This upper bounds the episode return to 1000 making it easier to compute aggregated performance measures across tasks. To facilitate fair wall-clock time comparison all algorithms are trained on the same hardware (i.e., a single NVIDIA V100 GPU machine) and evaluated with the same periodicity of 20000 environment steps. Each evaluation query averages episode returns over 10 episodes. Per common practice (Hafner et al., 2019), we employ action repeat of 2 and measure sample complexity in the environment steps, rather than the actor steps. In all the figures we plot the mean performance over 10 seeds together with the shaded regions which represent 95% confidence intervals. A full list of hyper-parameters can be found in Appendix E.

**Comparison Axes** In many real-world applications, taking a step in the environment incurs significant computational cost making sample efficiency a critical feature of an RL algorithm. It is hence important to compare RL algorithms in terms of their sample efficiency. We facilitate this comparison by computing an algorithm's performance measured by episode return with respect to environment steps. On the other end, striving low sample complexity often comes at the cost of a poor computational efficiency. Unfortunately, recent deep RL literature has paid very little attention to this important axis, which has led to skyrocketing hardware requirements. Such a trend has made it virtually impossible for an RL practitioner with modest hardware capacity to contribute to advancements in image-based RL, leaving research in this area to a few well-equipped labs. To democratize research in visual RL, we additionally propose to compare the agents in terms of wall-clock training time given the same single GPU hardware. We note that it is possible to adapt DrQ-v2 to a distributed setup, as has been done for DDPG in prior work (Barth-Maron et al., 2018; Hoffman et al., 2020).

### 4.2 COMPARISON TO MODEL-FREE METHODS

**Baselines** We compare our method to several state-of-the-art model-free algorithms for visual RL including CURL (Srinivas et al., 2020), DrQ (Yarats et al., 2021b), and vanilla SAC (Haarnoja et al., 2018a) augmented with the convolutional encoder from SAC-AE (Yarats et al., 2019). Vanilla SAC is a weak baseline and only included as a ground point to showcase the recent progress in visual RL.

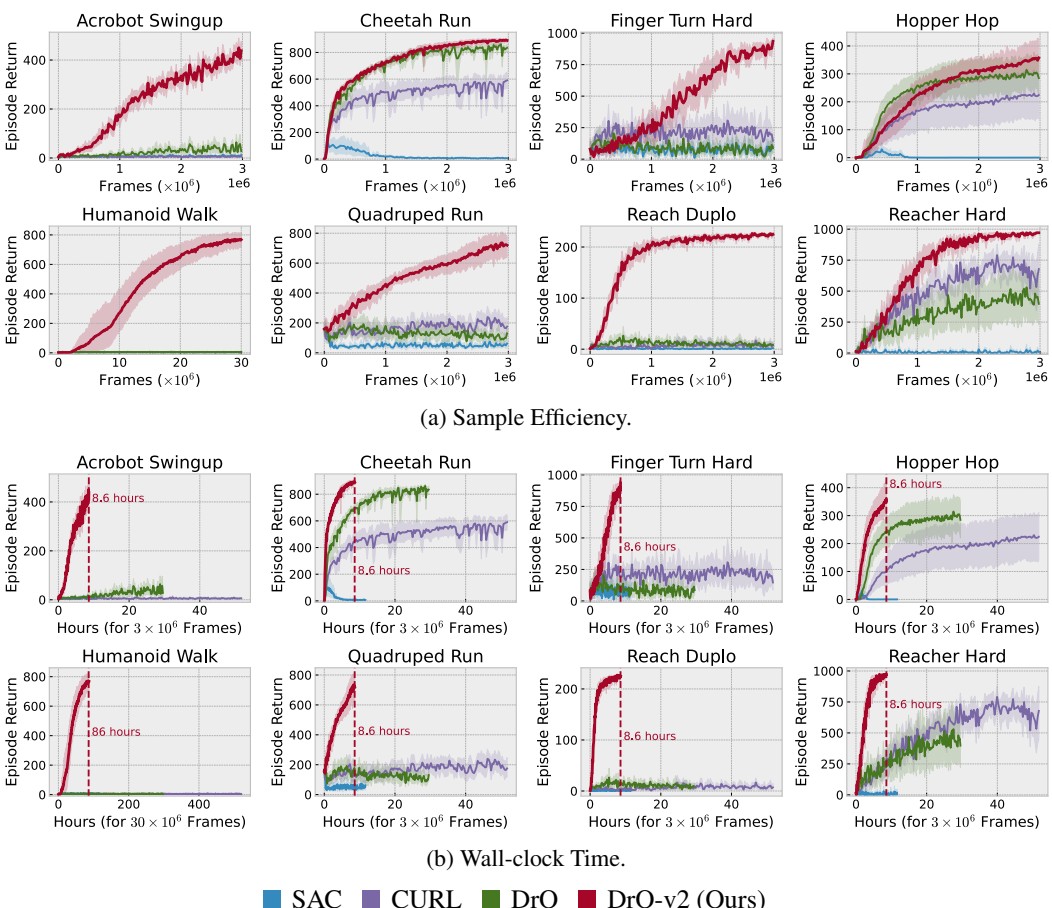

Figure 3: We compare DrQ-v2 on a subset of continuous control tasks that offer various challenges, including complex dynamics, sparse rewards, hard exploration, and more. (**a**) DrQ-v2 demonstrates favorable sample efficiency and comfortably outperforms leading model-free baselines, as well as requiring less wall-clock training image (**b**).

**Sample Efficiency Axis**   We present results on several *medium* and *hard* tasks in Figure 3a. Full results can be found in Appendix (Figure 6, Figure 8, and Figure 10). Our empirical study reveals that DrQ-v2 outperforms prior model-free methods in terms of sample efficiency across the three benchmarks with different levels of difficulty. Importantly, DrQ-v2's advantage is more pronounced on harder tasks (i.e., acrobot, quadruped, and humanoid), where exploration is especially challenging. Finally, DrQ-v2 solves the DMC humanoid locomotion tasks directly from pixels, which, to the best of our knowledge, is the first successful demonstration of such feat by a model-free method.

**Compute Efficiency Axis**   To facilitate a fair comparison in terms of sheer wall-clock training time, besides employee the identical training protocol (see Section 4.1), we also use the same mini-batch size of 256 for each agent. In Figure 13, we evaluate DrQ-v2 on a subset of DMC tasks for the sake of brevity only, and note that the demonstrated results can be easily extrapolated to the other tasks given the linear dependency between training time and sample complexity. In our benchmarks, DrQ-v2 is able to achieve a throughput of 96 FPS, which favorably compares to DrQ's 28 FPS (a $3.4\times$ increase), and CURL's 16 FPS (a $6\times$ increase) throughputs. Practically, DrQ-v2 solves easy, medium, and hard tasks within 2.9, 8.6, and 86 hours respectively. Full results can be found in Appendix (Figure 7, Figure 9, and Figure 11).

## 4.3    COMPARISON TO MODEL-BASED METHODS

**Baseline**   To see how DrQ-v2 stacks up against model-based methods, which tend to achieve better sample complexity in expense of a larger computational footprint, we also compare to recent and

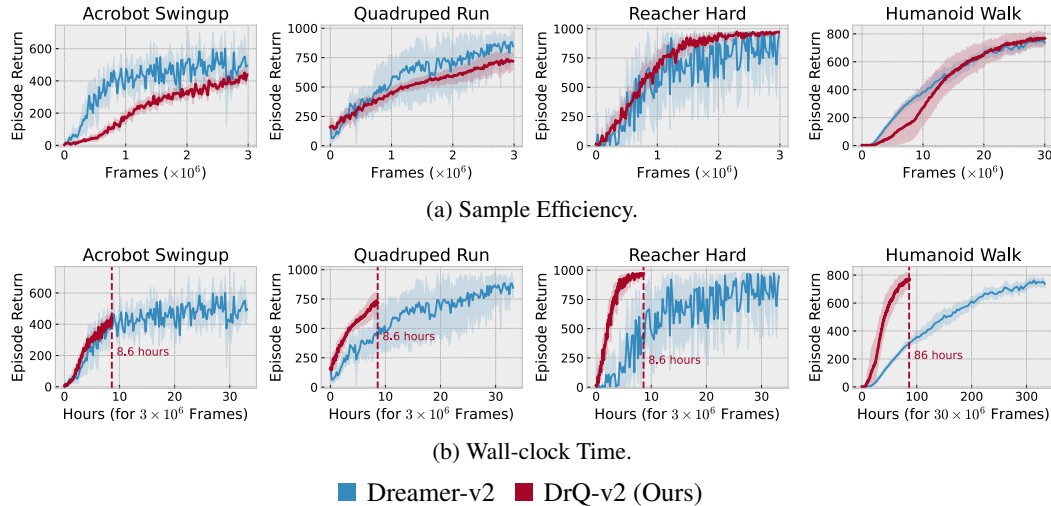

(a) Sample Efficiency.

(b) Wall-clock Time.

■ Dreamer-v2  ■ DrQ-v2 (Ours)

Figure 4: Model-based Dreamer-v2 needs to train a world model and thus performs more computations during training than model-free DrQ-v2. Still, (**a**) DrQ-v2 is able to match Dreamer-v2's sample efficiency, while (**b**) requiring much less wall-clock training time.

unpublished[2] improvements to Dreamer-v2 (Hafner et al., 2020), a leading model-based approach for visual continuous control. The recent update shows that the model-based approach can solve the DMC humanoid tasks directly from pixel inputs. The open-source implementation of Dreamer-v2 (https://github.com/danijar/dreamerv2) only provides learning curves for *Humanoid Walk*. For this reason we run their code to obtain results on other DMC tasks. To limit hardware requirements of compute-expensive Dreamer-v2, we only run it on a subset of 12 out of 24 considered tasks. This subset, however, overlaps with all the three (i.e. easy, medium, and hard) benchmarks.

**Sample Efficiency Axis**  Our empirical study in Figure 4a reveals that in many cases, DrQ-v2, despite being a model-free method, can rival sample efficiency of state-of-the-art model-based Dreamer-v2. We note, however, that on several tasks (for example *Acrobot Swingup*) Dreamer-v2 outperforms DrQ-v2. We leave investigation of such discrepancy for future work. Full results are provided in Appendix D (Figure 12).

**Compute Efficiency Axis**  A different picture emerges if comparison is done with respect to wall-clock training time. Dreamer-v2, being a model-based method, performs significantly more floating point operations to reach its sample efficiency. In our benchmarks, Dreamer-v2 records a throughput of 24 FPS, which is 4× less than DrQ-v2's throughput of 96 FPS, measured on the same hardware. In Figure 4b we plot learning curves against wall-clock time and observe that DrQ-v2 takes less time to solve the tasks. Full results can be found in Appendix (Figure 13).

## 4.4 ABLATION STUDY

In this section we present an extensive ablation study that guided us to the final version of DrQ-v2. Here, for brevity we only discuss experiments that were most impactful and omit others that did not pan out. For computational reasons, we only ablate on 3 different control tasks of various difficulty levels. Our findings are summarized in Figure 5 and detailed below.

**Switching from SAC to DDPG**  DrQ (Yarats et al., 2021b) leverages SAC (Haarnoja et al., 2018a) as the backbone RL algorithm. While it has been demonstrated by many works, including the original manuscripts (Haarnoja et al., 2018a;b) that SAC is superior to DDPG (Lillicrap et al., 2015b), our careful examination identifies two shortcomings that preclude SAC (within DrQ) to solve hard exploration-wise image-based tasks. First, the automatic entropy adjustment strategy, introduced in Haarnoja et al. (2018b), is inadequate and in some cases leads to a premature entropy collapse.

---

[2]ArXiv v3 revision from May 3, 2021 introduces a new result on the *Humanoid Walk* task in Appendix D.

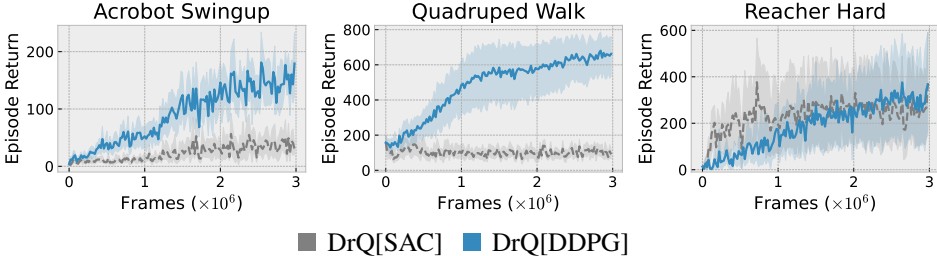

(a) DrQ (dotted silver) relies on SAC as a base RL algorithm. Replacing SAC with DDPG results in a significant performance gain (blue).

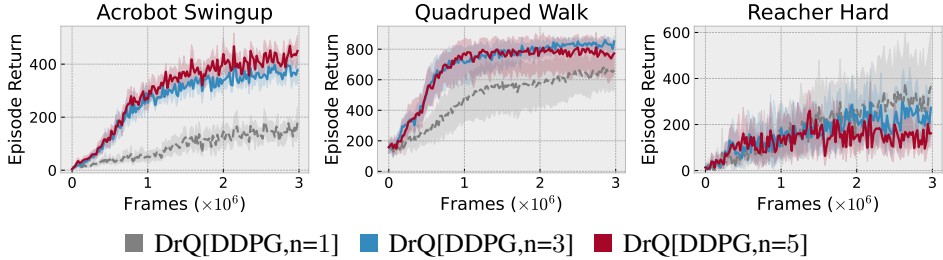

(b) DDPG straightforwardly incorporates $n$-step returns, a critical tool for exploration. We observe that the 3 (blue) and 5 (red) steps variants provide additional improvements to the previous version that uses single step TD-targets (silver). Going forward, we adopt 3-step returns (blue).

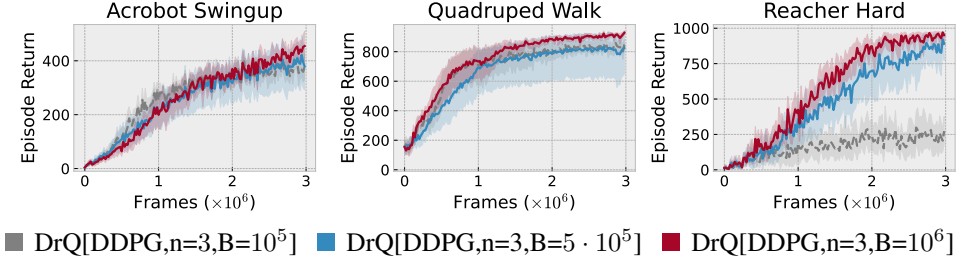

(c) Increasing the size of the replay buffer (B) improves performance, over the original $10^5$ used by DrQ (silver). Going forward, we use a buffer size of $10^6$ (red).

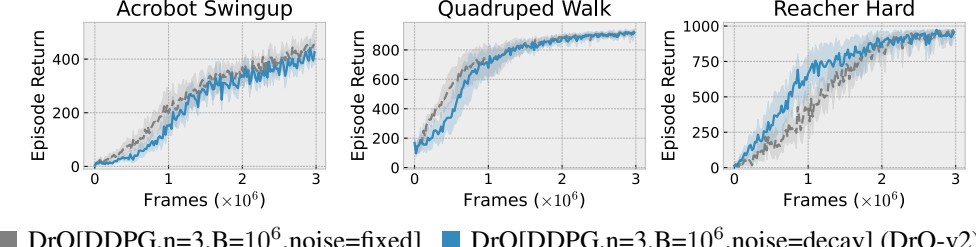

(d) Finally, a decaying schedule for the variance of the exploration noise (blue) helps on hard exploration tasks, versus the fixed variance variant (silver).

Figure 5: An ablation study that led us to the final version of DrQ-v2. We incrementally show each of the four key improvements to DrQ that collectively form DrQ-v2. The silver dotted curves in the first row show the original DrQ. In subsequent rows they show progressive improvements, using the optimal choice from the previous rows (i.e., the silver curve in the third row shows DrQ with a DDPG base RL algorithm and 3-step returns). The red and blue curves show the effect of individual modifications. In the last row the blue curve corresponds to DrQ-v2.

This prevents the agent from finding more optimal behaviors due to the insufficient exploration. In Figure 5a, we empirically verify our intuition and, indeed, observe that DDPG demonstrates better exploration properties than SAC. Here, DDPG uses constant $\sigma = 0.2$ for the exploration noise.

**N-step Returns**  The second issue concerns the inability of soft Q-learning to incorporate $n$-step returns to estimate TD error in a straightforward manner. The reason for this is that computing a target value for soft Q-function requires estimating per-step entropy of the policy, which is challenging to do for large $n$ in the off-policy regime. In contrast, DDPG does not require estimating per-step entropy to compute targets and is more amenable for $n$-step returns. In Figure 5b we demonstrate that estimating TD error with $n$-step returns improves sample efficiency over vanilla DDPG. We select 3-step returns as a sensible choice for our method.

**Replay Buffer Size**  We hypothesize that a larger replay buffer plays an important role in circumventing the catastrophic forgetting problem (Fedus et al., 2020). This issue is especially prominent in tasks with more diverse initial state distributions (i.e., reacher or humanoid tasks), where the vast variety of possible behaviors requires significantly larger memory. We confirm this intuition by ablating the size of the replay buffer in Figure 5c, where we observe that a buffer size of 1M helps to improve performance on *Reacher Hard* considerably.

**Scheduled Exploration Noise**  Finally, we demonstrate that it is useful to decay the variance of the exploration noise over the course of training according to Equation (3). In Figure 5d, we compare two versions of our algorithm, where the first variant uses a fixed standard deviation of $\sigma = 0.2$, while the second variant employes the decaying schedule $\sigma(t)$, with parameters $\sigma_{\text{init}} = 1.0$, $\sigma_{\text{final}} = 0.1$, and $T = 500000$. Having the exploration noise to decay linearly over time turns out to be helpful and provide an additional performance boost, which was especially useful for solving humanoid tasks.

## 5  RELATED WORK

**Visual Reinforcement Learning**  Successes of visual representation learning in computer vision (Vincent et al., 2008; Doersch et al., 2015; Wang and Gupta, 2015; Noroozi and Favaro, 2016; Zhang et al., 2017; Gidaris et al., 2018) has inspired successes in visual RL, where coherent representations are learned alongside RL. Works such as SAC-AE (Yarats et al., 2019), PlaNet (Hafner et al., 2018), and SLAC (Lee et al., 2019), demonstrated how auto-encoders (Finn et al., 2015) could improve visual RL. Following this, other self-supervised objectives such as contrastive learning in CURL (Srinivas et al., 2020) and ATC (Stooke et al., 2020), self-prediction in SPR (Schwarzer et al., 2020a), contrastive cluster assignment in Proto-RL (Yarats et al., 2021a), and augmented data in DrQ (Yarats et al., 2021b) and RAD (Laskin et al., 2020), have significantly bridged the gap between state-based and image-based RL. Future prediction objectives (Hafner et al., 2018; 2019; Yan et al., 2020; Finn et al., 2015; Pinto et al., 2016; Agrawal et al., 2016) and other auxiliary objectives (Jaderberg et al., 2016; Zhan et al., 2020; Young et al., 2020; Chen et al., 2020) have shown improvements on a variety of problems ranging from gameplay, continuous control, and robotics. In the context of visual control settings, clever use of augmented data (Yarats et al., 2021b; Laskin et al., 2020) currently produces state-of-the-art results on visual tasks from DMC (Tassa et al., 2018).

**Humanoid Control**  The humanoid control problem first presented in Tassa et al. (2012), has been studied as one of the hardest control problems due to its large state and action spaces. The earliest solutions to this problem use ideas in model-based optimal control to generate policies given an accurate model of the humanoid . Subsequent works in RL have shown that model-free policies can solve the humanoid control problem given access to proprioceptive state observations. However, solving such a problem from visual observations has been a challenging problem, with leading RL algorithms making little progress to solve the task (Tassa et al., 2018). Recently, Hafner et al. (2020) was able to solve this problem through a model-based technique in around 30M environment steps and 340 hours of training on a single GPU machine. DrQ-v2, presented in this paper, marks the first model-free RL method that can solve humanoid control from visual observations, taking also around 30M steps and 86 hours of training on the same hardware.

## 6  CONCLUSION

We have introduced a conceptually simple model-free actor-critic RL agent for image-based continuous control – DrQ-v2. Our method provides significantly better computational footprint and masters tasks from DMC directly from pixels, most notably the humanoid locomotion tasks that

were previously unsolved by model-free approaches. To support our empirical results and inspire further research in visual RL we provide an efficient PyTorch implementation of DrQ-v2 at https://anonymous.4open.science/r/drqv2.

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
