# OpenReview forum: "Mastering Visual Continuous Control: Improved Data-Augmented Reinforcement Learning"
_ICLR.cc/2022/Conference — ICLR 2022 Poster_

### Official Review · Reviewer_K4wT · 2021-10-25

**Correctness:** 4
**Technical Novelty And Significance:** 1
**Empirical Novelty And Significance:** 3
**Recommendation:** 8
**Confidence:** 3

**Main Review:**

Pro:
1. An improved implementation to decrease the training time for visual-based RL. And thanks for pointing out the importance of comparing wall clock time, which is usually neglected in related work.
2. Solving the humanoid task, which is not possible with prior implementation/method.
3. The approach is well documented with open source code.

Issue
1. . "First, the automatic entropy adjustment strategy, introduced in Haarnoja et al. (2018a), is inadequate and in some cases leads to a premature entropy collapse." What about don't use the automatic entropy adjustment? By the way, the reference to Haarnoja et al is duplicated. Since the whole point of SAC is to encourage exploration with the additional entropy reward, it is interesting to see that the authors make the comment that "DDPG demonstrates better exploration properties than SAC". It will be great if this point can be investigated further.
2. It is interesting that the buffer size for quadruped run is 10^5. It will be nice if the authors can provide some insights on why this is an exception. What happens if I set it to be the default 10^6? When and how should I tune this parameter? Wouldn't this contradict the result of Figure 5(c)?
3. It is not obvious to me that a decaying schedule helps from Figure 5(d). Even for reacher hard, eventually the fixed variance curve reaches the optimal around the same time as schedule decay.

Additional comments:
1. From the perspective of simulation only baseline, 96FPS is really slow. One would wonder if faster training time can be achieved with on policy algorithm such as PPO, which can leverage parallel simulations and also less gradient update per sample collected. Of course, I understand this is not in the scope of this paper.

**Summary Of The Paper:**

This paper presents DrQ-v2,  an improvement over DrQ, for solving visual-based RL problems. Key components include (1) DDPG over SAC (2) n-step returns for the critic (3) replay buffer size and (4) decaying exploration. Implementation improvement is also discussed. Notably, DrQ-v2 is able to solve the humanoid task effectively compared to recent work in this domain.

**Summary Of The Review:**

This paper presents various improvements for visual RL with continuous control. Various ablations support most of the claims in the paper and notably the approach presented can solve the humanoid task that is not possible with prior methods.
One interesting aspect of the paper is that it provides different perspectives on RL benchmark. e.g., more emphasis on wall clock time, DDPG is actually better than SAC in the visual domain.
If my concern in the main review is addressed appropriately, I believe this will be valuable to the community.

---

> ### Author Response · Authors · 2021-11-11
> **Rebuttal**
>
> Thank you for your time and constructive feedback on DrQ-v2. We’re glad that you find our method to be simple and efficient. We also appreciate you recognizing that our method is able to solve previously unsolved humanoid tasks. We address your questions below.
>
> **Automatic entropy adjustment strategy**:
> Indeed, that was actually the very first thing we decided to try when we noticed the problem of early entropy collapsing when the automatic entropy adjustment strategy is used in SAC. We tried to keep alpha fixed, as was proposed in the original SAC paper, but the results were mixed. More importantly the fixed alpha variant didn’t perform much better on harder tasks (see this result: https://imgur.com/dcX8JcK), despite extensive tuning of alpha. That was one of the main reasons (besides n-step returns) that led us to try DDPG. We also note the followup SAC paper (https://arxiv.org/pdf/1812.05905.pdf) echoes our findings and in Figure 1 it shows that learned temperature performs slightly better than fixed temperature. We suspect that in harder tasks (especially tasks with sparse rewards) adding entropy bonus introduces bias that affects learning, as a different objective is being optimized.  Hopefully this datapoint addresses your concern. We have also fixed the duplicate reference problem and uploaded the updated manuscript.
>
> **Buffer size.**
> The smaller buffer size for Quadruped Run was guided by empirical evidence. In our experiments we noticed that Quadruped Run is the only task that benefits from a smaller replay buffer: https://imgur.com/3frwDt8. Still DrQ-v2 is able to learn a good policy with the 1M replay buffer and outperform prior methods.
>
> **Decay schedule**:
> We agree that the tasks in Figure 5 (d) do not justify the usage of the decaying schedule. However, we noticed that the decaying schedule is very important in the humanoid tasks, see this figure: https://imgur.com/OGSuFMd. We decided to not show this result in order to keep the same set of tasks for all the subfigures in Figure 5, which in retrospect, might not have been the ideal choice. We will update the paper with this additional result. Hopefully this addresses your concern.
>
>
> **Parallel simulation**:
> Both DrQ-v2 and PPO can benefit from a parallel/distributed implementation, as both on-policy and off-policy algorithms are amenable to such modifications (see for example D4PG paper where a distributed version is considered). We thus believe that DrQ-v2’s FPS can be greatly improved with linear scaling. We decided to focus on a single GPU setup as one of the goals of our work was to produce a simple and fast algorithm that can be run on modest hardware and be adopted by academic labs.
>
>
> In the light of the above clarifications, we would like to ask if you are willing to increase your score assuming we have addressed your concerns. Otherwise, please let us know if you have additional questions.

---

> > ### Comment · Reviewer_K4wT · 2021-11-16
> > **socre updated**
> >
> > Thanks for your clarification. I have updated my score accordingly.

---

### Official Review · Reviewer_7c6j · 2021-10-29

**Correctness:** 3
**Technical Novelty And Significance:** 1
**Empirical Novelty And Significance:** 3
**Recommendation:** 8
**Confidence:** 5

**Main Review:**

##########################################################################

Summary:

This paper introduces DrQ-v2, an improvement over DrQ, a model free off-policy actor-critic approach which uses SAC+data augmentation to learn directly from pixels.
DrQ-v2, on the other hand, switches SAC with DDPG and proposes a series of algorithmic, hyperparameter choices and implementation improvements.
These improvements make it both faster computationally and better empirically as it is able to solve the Humanoid environment from pixel.


##########################################################################

Pros:

- The paper solves the humanoid environment from visual input, a task long overdue in the RL community, with an easy to implement and easy to understand algorithm.

- To solve this task, using this algorithm, it is not required to have inaccessible hardware requirements.

- The paper provides open-source implementation, pseudo-code, well designed and comprehensive experiments for the improvements proposed over DrQ, which are run over 10 seeds to provide more claims over reproducibility.

##########################################################################

Cons:

- With the adding of clipped double Q-learning and action noise, you are technically switching from SAC to TD3 more than DDPG. This could be at least highlighted in the last paragraph of the introduction where the algorithmic changes are summarized.

- Although the proposed paper provides several ablation studies and extensive experiments, I still think the following experiments would be nice to have. From my understanding the improvements are:
    a. swith sac to ddpg with clipped double Q-learning and action noise
    b. multi step return
    c. exploration schedule
    d. adding bilinear interpolation to random shift of image
    e. replay buffer size, learning rate and batch size hyperparameter
    f. speed improvements provided by better implementations of replay buffer and data augmentation.

1. Are you sure that the wall-clock time improvements are related to the algorithm and not to your better implementation of replay buffer and data augmentation (f)? Have you tried to run DrQ with the new implementations of replay buffer and data augmentation?

2. The improvements above could be grouped into algorithmic improvements allowed by switching backbone to ddpg (a, b, c), and code implementation/hyperparameter improvements (d, e, f). For qualifying the importance of the algorithmic improvements, a stronger DrQ baseline could be introduced with the addition of (d), (e), (f) to DrQ and compared to DrQ-v2.

##########################################################################

Questions during rebuttal period:

Address and/or clarify the cons above.

##########################################################################

Reasons for score:

Overall I am towards acceptance of the paper. I think it's straightforward to read and the claims are well documented and backed by empirical results. There are no real technical contributions but the empirical novelties of changing to DDPG/TD3 and the performances are good. Half of the contributions indicated are just related to hyperparameter tuning and optimized implementations of the same algorithm and for this reason it's not a higher score. Hopefully the authors can address my concern in the rebuttal period.


**Summary Of The Paper:**

This paper introduces DrQ-v2, an improvement over DrQ, a model free off-policy actor-critic approach which uses SAC+data augmentation to learn directly from pixels.
DrQ-v2, on the other hand, switches SAC with DDPG and proposes a series of algorithmic, hyperparameter choices and implementation improvements.
These improvements make it both faster computationally and better empirically as it is able to solve the Humanoid environment from pixel.

**Summary Of The Review:**

Overall I am towards acceptance of the paper. I think it's straightforward to read and the claims are well documented and backed by empirical results. There are no real technical contributions but the empirical novelties of changing to DDPG/TD3 and the performances are good. Half of the contributions indicated are just related to hyperparameter tuning and optimized implementations of the same algorithm and for this reason it's not a higher score. Hopefully the authors can address my concern in the rebuttal period.

---

> ### Author Response · Authors · 2021-11-10
> **Rebuttal**
>
> Thank you for your time and encouraging feedback on DrQ-v2. We’re glad that you agree with our excitement about our method being able to solve long standing humanoid tasks!  We answer your questions below.
>
> **More TD3 than DDPG**:
> We consider our algorithm to be a variant that stands in between DDPG and TD3, but we thought it is still closer to DDPG rather than TD3. TD3 adds four components on top of DDPG: 1) clipped double Q-learning, 2) delayed policy update, 3) target policy, 4) policy smoothing regularization. We only use 1) and introduce our own decaying schedule to replace 4). That is why we decided to use DDPG as a name, rather than TD3, we however not that 1) is a very important component of DrQ-v2 and we cite it appropriately. Still, we have updated the paragraph in the introduction that you mentioned to summarize our algorithmic changes better.
>
> **Algorithmic and implementation improvements**:
> We agree with your assessment that our contributions can be grouped into algorithmic (a, b, c) and implementation (d, e, f) categories. To answer your question we have conducted the requested experiment here: https://imgur.com/HHLCz9h. We summarize our findings:
> 1) Adding changes (d, e) does improve sample efficiency of DrQ, but it is still far away from DrQ-v2's sample efficiency. For example, the modified DrQ is still unable to make any progress on Humanoid Walk.
> 2) Adding the change (f) indeed improves wall-clock training time of DrQ and gets it closer to training time of DrQ-v2. However, there is still some delta due to the fact that DrQ-v2 requires updating the critic network 2x less often than DrQ without loss in performance. This is possible due to DDPG and n-step returns.
>
> We hope this additional experiment addresses your concerns.
>
>
>
> In the light of the above clarifications, we would like to ask if you are willing to increase your score assuming we have addressed your concerns. Otherwise, please let us know if you have additional questions.

---

> > ### Comment · Reviewer_7c6j · 2021-11-17
> > **Score updated.**
> >
> > Thanks for your comment, I updated my score. It seems like the main contribution of this paper, more than the hyperparameters and the better code implementation, is indeed the realization that DDPG/TD3 allows for faster learning compared to SAC.

---

### Official Review · Reviewer_uMh5 · 2021-11-02

**Correctness:** 3
**Technical Novelty And Significance:** 2
**Empirical Novelty And Significance:** 2
**Recommendation:** 6
**Confidence:** 5

**Main Review:**

**Strengths**:

- This method is simple and efficient.

**Weaknesses**:

- The generalization of the DrQ-v2 method on different tasks and RL algorithms:
  - Generalization on different tasks: How DrQ-v2 performs on Atari games? DreamerV2 can *master* Atari. How about DrQ-v2?
  - Generalization on RL algorithms: How DrQ-v2 performs when incorporated with on-policy algorithms, such as TRPO and PPO? Can DrQ-v2-PPO achieve competitive or better performance?


**Summary Of The Paper:**

This paper proposed a new model-free RL method for visual continuous control problems.

**Summary Of The Review:**

This paper proposed a new model-free RL method for visual continuous control problems. This method is efficient and straightforward. But the evaluation of the proposed method is limited. If the author can conduct more experiments on Atari games and with on-policy algorithms (such as PPO) can further improve the value of this work.

---

> ### Author Response · Authors · 2021-11-11
> **Rebuttal**
>
> Thank you for your time and feedback on DrQ-v2. We’re glad that you find our method to be simple and efficient. We address each of your questions below.
>
> **Generalization to different tasks, such as Atari**:
> Although this paper tackles the DM Control benchmark on **24 tasks** total tasks, which is the one of the most extensive comparisons to date, the insights from this work are readily transferable to other domains. To provide context, ideas of using augmentations for faster RL that first originated from DrQ has already improved robotic performance [FERM, Zhan & Zhao et al., 2020], Atari games [SPR, Schwarzer et al., 2020; DrQ, Yarats et al. 2021], and Procgen [DrAC, Raileanu et al., 2020]. Since DrQ-v2 accelerates data augmentation based learning, we believe the algorithm should readily transfer to other domains. With respect to the experiments DrQ-v2 already goes beyond most of the prior work in terms of the scale of empirical study. We want to emphasize that the current experiments already resulted in a large computational cost (24 tasks x 5 algorithms x 10 seeds = 2400 total runs just for the final results, without ablation). Adding additional domains, such as Atari, which consists of 60 tasks each needed to train for 200M frames and 5 seeds, would result in unjustifiable computation and unnecessary CO2 emission.
>
> **Generalization to different RL algorithms**:
> The techniques from the original DrQ paper were adapted to different algorithms, such as PPO [DrAC, Raileanu et al., 2020], Dreamer [Dreaming, Okada et al., 2020], Rainbow [SPR,  Schwarzer et al., 2020]. Given that DrQ-v2 bears a lot of similarities to DrQ, we believe that techniques from DrQ-v2 can be applied to other algorithms as well. We note that our empirical study was already extremely extensive in terms of number of tasks and baselines, and adding additional baselines wasn't feasible from a computational point of view.
>
> In the light of the above clarifications, we would like to ask if you are willing to increase your score assuming we have addressed your concerns. Otherwise, please let us know if you have additional questions.

---

### Official Review · Reviewer_ncsH · 2021-11-08

**Correctness:** 1
**Technical Novelty And Significance:** 2
**Empirical Novelty And Significance:** 2
**Recommendation:** 5
**Confidence:** 5

**Main Review:**

The paper is well written and easy to read. Results are well explained, ablated and compared with the current state of the art. The improvements in terms of sample efficiency and speed (on a single V100 GPU) are a step change with respect to results which have been published before.

The main limitation of the paper is its limited scope which significantly reduces the technical novelty and significance to the field. The paper  isn't substantially presenting anything new neither from the theoretical or empirical standpoint. The main contribution consists of a combination of previously existing techniques carefully optimized together. It isn't clear how the proposed sample efficiency and speed improvements could generalize beyond the DM control suite without further adjustments. In a sense, authors don't do any effort to prove that their improvements could go beyond the scope presented in the paper and this limits the significance to the field.

**Summary Of The Paper:**

The paper presents DrQ-v2, a  model-free reinforcement learning (RL) algorithm for visual continuous control. The algorithm is tested on the Deepmind Control Suite and it is proven to learn visual tasks in a remarkably short wall-clock time and with an extremely high computational efficiency (i.e. small number of frames). The paper is an improvement over a previous algorithm DrQ. Improvements are achieved with: (1) a new learning backbone (DDPG vs SAC); (2) the introduction of n-steps returns; (3) a bigger reply buffer and (4) a better scheduled exploration strategy.

**Summary Of The Review:**

The paper is overall a good paper but I think it's marginally below the acceptance threshold because of its limited technical novelty and significance. It is worth of a publication as a technical report but it has a marginal value for the audience of an international conference.

---

> ### Author Response · Authors · 2021-11-10
> **Rebuttal**
>
> Thank you for your time and feedback on DrQ-v2. We’re glad that you agree that our work introduces a *step increase* in performance on DMControl tasks compared to prior work and that our paper is well written and easy to read. Given your reservations on our paper’s novelty and significance we would like to address it in detail below.
>
>
> **Significance in the RL community**:
> For any field to move forward it needs relevant benchmarks (ImageNet in CV, Google 1 Billion Word Corpus in NLP, etc.) and algorithms that improve these benchmarks. DrQ-v2 presents an approach that demonstrates a *step increase* in performance on DMControl (a leading benchmark in image-based RL) and requires 3.5x less wall-clock training time than prior work. Additionally, DrQ-v2 is the first algorithm that solves humanoid directly from pixels, which is significant and echoed by **reviewer 7c6j**: *”The paper solves the humanoid environment from visual input, a task long overdue in the RL community”* and **reviewer K4wT**: *”Solving the humanoid task, which is not possible with prior implementation/method.”*.
> We thus believe that our work is an important contribution to the RL community as it provides a reliable and reproducible algorithm with a non-trivial delta in sample and time efficiency. To further emphasize the potential impact of DrQ-v2, the previous version of this algorithm DrQ has already had significant impact in RL with >100 papers having used it in the last year. Since DrQ-v2 performs better and is 3.5x faster than DrQ, we believe DrQ-v2 will not only serve as a new backbone for visual RL, but also due to its relatively smaller computational footprint it will make research in RL more accessible to the academic community.
>
> **New empirical standpoint**:
> We disagree with the characterization that this work does not present anything new from an empirical standpoint. Several new empirical insights have been revealed in this work through careful ablation and large-scale experimentation. Some of these are listed below:
> 1) Demonstrating that DDPG performs better than SAC. This goes against common knowledge in the RL community and is an important result by itself.
> 2) Importance of n-step returns, which was overlooked in image-based continuous control, mostly because SAC doesn’t allow incorporating it straightforwardly.
>
>
>
> **Beyond the scope of DM Control**:
> Although this paper tackles the DM Control benchmark on **24 tasks** total tasks, which is the one of the most extensive comparisons to date, the insights from this work are readily transferable to other domains. To provide context, ideas of using augmentations for faster RL that first originated from DrQ has already improved robotic performance [Zhan & Zhao et al., 2020], Atari games [Yarats et al. 2021], and Procgen [Raileanu et al., 2020]. Since DrQ-v2 accelerates data augmentation based learning, we believe the algorithm should readily transfer to other domains. With respect to the experiments DrQ-v2 already goes beyond most of the prior work in terms of the scale of empirical study. We want to emphasize that the current experiments already resulted in a large computational cost (24 tasks x 5 algorithms x 10 seeds = 2400 total runs just for the final results, without ablation). Adding additional domains, such as Atari, which consists of 60 tasks each needed to train for 200M frames and 5 seeds, would result in unjustifiable computational footprint and unnecessary CO2 emission.
>
>
> In the light of the above clarifications, we would like to ask you to reconsider your initial assessment regarding the significance and novelty of our work, and increase the score accordingly if you think that we have addressed your concerns. Otherwise, please provide us with more actionable questions.

---

> > ### Comment · Reviewer_ncsH · 2021-11-22
> > **Thanks for the rebuttal**
> >
> > Thanks, the authors raised good points but they don't justify any change to my initial assessment.

---

### Public Comment · ~Paul_Weng1 · 2021-11-11
**Questions about image augmentation**

Interesting improvements compared to "Image augmentation is all you need"!

Regarding the image augmentation part, have you evaluated the performance gain when:
- using or not the bilinear interpolation?
- using only transformed observations vs using original observation+transformed observation?

---

### Decision · Program_Chairs · 2022-01-20

**Decision:**

Accept (Poster)

**Comment:**

The paper addresses various improvements in visual continuous RL, based on a previous RL algorithm (DrQ). As the reviewers point out, the main contribution of the paper is of empirical nature, demonstrating how several different choices relative to DrQ significantly improve data efficiency and wall-clock computation, such that several control problems of the DeepMind control suite can be solved more efficiently. The average rating for the paper is above the acceptance threshold, and some reviewers increased their rating after there rebuttal. While a mostly empirically motivated papers is always a bit more controversial, the paper may nevertheless stimulated an interesting discussion at ICLR that will be beneficial for the community, and should thus be accepted.